# Spatio-Temporal Niche of Sympatric Tufted Deer (*Elaphodus cephalophus*) and Sambar (*Rusa unicolor*) Based on Camera Traps in the Gongga Mountain National Nature Reserve, China

**DOI:** 10.3390/ani12192694

**Published:** 2022-10-07

**Authors:** Zhiyuan You, Bigeng Lu, Beibei Du, Wei Liu, Yong Jiang, Guangfa Ruan, Nan Yang

**Affiliations:** 1Institute of Qinghai-Tibetan Plateau, Southwest Minzu University, Chengdu 610041, China; 2Zoololical Society of Sichuan Province, Chengdu 610065, China; 3School of Animal and Veterinary Sciences, Southwest Minzu University, Chengdu 610041, China; 4Gongga Mountain National Nature Reserve Adiministration Bureau, Kangding 626000, China

**Keywords:** camera trap, tufted deer, sambar, sympatric relative species, spatio-temporal niche

## Abstract

**Simple Summary:**

The broad adoption of camera traps in wildlife monitoring leads to a better understanding of the distribution and activity patterns of wild animals. In this study, two endangered wild animals distributed in the Gongga Mountain National Nature Reserve, the tufted deer (*Elaphodus cephalophus*) and sambar (*Rusa unicolor*), were investigated for habitat suitability and rhythm activity patterns based on 9-year (2012–2021) camera-trap data. The results revealed that the suitable range of the major environmental factors and the distribution range of tufted deer in the reserve were greater than those of sambar. The daily activity peaks in tufted deer and sambar were both at dusk and dawn, but sambar stayed active during the evening.

**Abstract:**

Clarifying the distribution pattern and overlapping relationship of sympatric relative species in the spatio-temporal niche is of great significance to the basic theory of community ecology and integrated management of multi-species habitats in the same landscape. In this study, based on a 9-year dataset (2012–2021) from 493 camera-trap sites in the Gongga Mountain National Nature Reserve, we analyzed the habitat distributions and activity patterns of tufted deer (*Elaphodus cephalophus*) and sambar (*Rusa unicolor*). (1) Combined with 235 and 153 valid presence sites of tufted deer and sambar, the MaxEnt model was used to analyze the distribution of the two species based on 11 ecological factors. The distribution areas of the two species were 1038.40 km^2^ and 692.67 km^2^, respectively, with an overlapping area of 656.67 km^2^. Additionally, the overlap indexes Schoener’s D (D) and Hellinger’s-based I (I) were 0.703 and 0.930, respectively. (2) Based on 10,437 and 5203 independent captures of tufted deer and sambar, their daily activity rhythms were calculated by using the kernel density estimation. The results showed that the daily activity peak in the two species appeared at dawn and dusk; however, the activity peak in tufted deer at dawn and dusk was later and earlier than sambar, respectively. Our findings revealed the spatio-temporal niche relationship between tufted deer and sambar, contributing to a further understanding of the coexistence mechanism and providing scientific information for effective wild animal conservation in the reserve and other areas in the southeastern edge of the Qinghai–Tibetan Plateau.

## 1. Introduction

Ecological niches primarily include temporal, spatial, and feeding dimensions [1,2]. In long-term natural evolution, each species tends to occupy a unique niche and minimize the degree of overlap with other species [3,4]. Sympatric relative species usually have similar ecological needs and complex interspecific relationships, and their niche differentiation has always been one hotspot of community ecology [5,6,7,8]. The study of sympatric relative species can not only improve the understanding of interspecific relationships and coexistence mechanisms but also enriches the basic theory of community ecology, which can help the administrators of protected areas to formulate proper conservation strategies [9,10]. 

Wild ungulates are one of the most important communities in forest and grassland ecosystems as they not only directly or indirectly affect the growth and renewal of plant communities by feeding, trampling, and carrying seeds but also are the main food source of carnivores, which can drive the composition and structure of these ecosystems [11,12,13]. In recent years, many scholars have carried out various studies on the niche differentiation and coexistence of sympatric ungulates in the wild along different niche dimensions. The study on the spatio-temporal niche patterns of seven sympatric wild ungulates distributed in the Qinling Mountains (China) provided a scientific reference of the spatial and temporal ecology of wild ungulate communities in forest ecosystems [14]. The trophic overlap in sympatric red deer (*Cervus elaphus*) and roe deer (*Capreolus capreolus*) in northeast China was very high, but these two species selected totally different habitats [15]. Bharal (*Pseudois nayaur*) in the Wanglang National Nature Reserve (Sichuan) showed different behavior patterns in their circadian rhythm when facing varied space and time [16].

Tufted deer and sambar belong to the family Cervidae of the order Artiodactyla and are the two most widely distributed deer species in China. They are listed as class II nationally key protected animals in China, and tufted deer are listed as Near Threatened (NT) and sambar as Vulnerable (VU) by the IUCN Red List [17]. The two species are herbivores in montane forests, playing important roles in maintaining the structural integrity and functional stability of ecosystems. There is a large overlap in the distribution area of these two species in China [17]; however, previous research mainly focuses on the selection and utilization of the habitat of one species, and no studies have yet revealed the temporal and spatial distribution pattern of the two species in any sympatry [18,19,20,21,22]. Based on the 9-year camera-trap data in the reserve, the spatio-temporal niche and the overlapping patterns between the two sympatric deer were studied in the Gongga Mountain National Nature Reserve, China. 

## 2. Materials and Methods

### 2.1. Study Area

Gongga Mountain National Nature Reserve (101°29′ to 102°12′ E, 29°01′ to 30°05′ N) is located in the middle of the Daxue Mountains of the Hengduan Mountains on the southeastern edge of the Qinghai–Tibetan Plateau, with a total area of approximately 4091.40 km^2^ (Figure 1). The reserve provides protection to the forest ecosystems of the Daxue Mountains (mainly the Gongga Mountains); various rare, wild animal and plant resources; and low-altitude modern glaciers. From bottom to top along the altitude gradient, it can be further divided into six climate zones: subtropical, warm, cold temperate, subarctic, cold, and ice–snow; moreover, along with the altitude, the vegetation shows a clear vertical structure [23,24,25]. There are 96 mammal species from 7 orders and 25 families distributed in the reserve, including 11 class I and 18 class II nationally key protected species in China [26]. Based on the periodic precipitation in the Gongga Mountains, early October to March is the dry season, and April to September is the rainy season in general.

### 2.2. Data Collection

#### 2.2.1. Species Presence Data

The species presence data were obtained from the monitoring records of 493 camera-trap sites in the reserve from 2012 to 2021, and the layout altitude range was 1930–4752 m. The infrared cameras were placed across the reserve at sites with higher presence probability of wild animals, such as trails used by wild animals, areas near water sources, and sites with animal feces or traces [27]. The camera was tied at a height of about 0.5 m from the ground with the lens parallel to the ground, and the obstacles before the camera were removed. The three infrared camera models used in the 9-year infrared camera monitoring project in the reserve were Yianws (L710-940) and Ltl Acorn (Ltl 6210, Ltl 6310). The total effective camera workday was 102879. All cameras were set to work the whole day with a mixed shooting mode (once triggered, taking 3 photos and then a 20 s video) and a medium or high sensitivity with no baits. 

To reduce the phenomenon of overfitting, the trim duplicate occurrences function of ENMTools 1.4 software was used to remove the duplicate presence points of the same species located in the same raster (30 m × 30 m) [28]. It resulted in only one randomly selected presence point for each species in each raster cell where the species occurred. 

#### 2.2.2. Environmental Data and Data Processing

Previous studies have shown that topographic, vegetation, climate, water source distance, and human disturbance variables are the main factors affecting the suitability of wildlife habitats [29,30,31]. Therefore, 26 ecological and environmental factors were selected to predict the suitable habitats of the two species, including: (1) Topographic variables, including elevation, slope, and aspect, were extracted from the DEM data layer (30 m resolution) of ASTER GDEM V3 in the Geospatial Data Cloud. (2) Vegetation variables included vegetation type and normalized difference vegetation index (NDVI). The vegetation type was derived from FROM-GLC version 2 of Tsinghua University Data Center and reclassified as 9 types (perennial/snowfield, alpine screes, alpine meadow, shrub, coniferous forest, broadleaved forest, mixed forest, water, and other). NDVI data were obtained from the annual maximum NDVI dataset in China from 2000 to 2020 (30 m resolution) from China National Ecosystem Science Data Center. Finally, the maximum NDVI of 2015 was selected to assure consistency with the other environmental data. (3) Climate variables, including 19 current bioclimatic factors under 1 km resolution, were obtained from the WorldClim database. To avoid the negative impact of high spatial collinearity among climate factors on the model prediction, the MaxEnt model and ENMTools 1.4 were used to screen out those factors with a contribution greater than 1% and a relative correlation value <0.75 [32]. (4) Water source distance and human disturbance variables, including distance to river and distance to human settlement, were obtained from the 1:250,000 basic geographic data of Sichuan Province (2015) from the China National Earth System Science Data Center (Table 1). Before the construction of the model, the minimum convex polygon was generated for the boundary of the reserve in ArcGIS 10.7. Finally, 11 environment variables, including elevation, slope, aspect, vegetation type, NDVI, isothermality, mean temperature of the driest quarter, total annual precipitation, precipitation seasonality, distance to river, and distance to human settlement, were resampled to 30 m resolution with FROM-GLC as the template then projected all layers to the coordinate system of WGS_1984_UTM_Zone_47N. 

### 2.3. Data Analysis

#### 2.3.1. MaxEnt Model

Features in MaxEnt are derived from environmental variables of 2 types: continuous and categorical. Continuous variables include isothermality, slope, mean temperature of the driest quarter, total annual precipitation, aspect, precipitation seasonality, altitude, NDVI, distance to river, and distance to settlement, and categorical variables are vegetation types. For each species, we randomly used 75% of occurrence points to train the MaxEnt mode, while the other 25% were used for the model test [33,34]. We replicated the model 10 times for each species using bootstrapping methods for model robustness. The MaxEnt models were outputted in logistic format, and the values in each raster were averaged across the 10 replicates to represent the habitat suitability index of each species. The habitat suitability index ranges from 0 to 1, with greater values indicating higher habitat suitability index [35]. The area under the ROC curve (AUC), which measures the quality of a ranking of sites, was used to evaluate the prediction. The AUC is the probability that a randomly chosen presence site will be ranked above a randomly chosen absence site. A random ranking has, on average, an AUC of 0.5, and a perfect ranking achieves the best possible AUC of 1.0. Models with values above 0.75 are considered potentially useful [36]. Meanwhile, response curves were used to analyze the contribution and effect of each environmental variable to species habitat suitability.

#### 2.3.2. Spatial Overlap Analysis 

Therefore, we used True Skill Statistic (TSS) as the threshold for dividing the habitat suitability of two species: TSS = sensitivity + specificity − 1 and ranges from −1 to 1, with 1 indicating a perfect fit and values less than 0 indicating a performance no better than random [37,38]. ArcGIS 10.7 software was used to binarize the prediction results of the tufted deer and sambar models. The areas with habitat suitability index > TSS were divided into potentially suitable areas; the areas with value < TSS were non-suitable areas. To quantify the similarity in distribution of the suitable habitats of tufted deer and sambar in the reserve, ENMTools1.4 software was used to calculate the niche overlap index Schoener’s D (D) and Hellinger’s-based I (I) of the two species based on the output results of MaxEnt. D and I were in the range of [0, 1], indicating that the spatial niches of the two species never overlapped to be completely consistent [28].

#### 2.3.3. Temporal Niche Analysis

Photos and videos were summarized by sites, hour, and date at each camera site. To ensure independence of photographic capture events, any consecutive photo or video of the same species within 30 min was recorded as an independent detection [39], and for all analyses, only independent detections were considered.

Kernel density estimation was used to compare the temporal niche overlaps in the two species. It was assumed that the activity density of the target species was proportional to the captured rate of the camera traps [40]. This method considers that each detection of a species is a random sample collected from a continuous daily activity rhythm distribution, which describes the probability of the species being detected in a certain period of time. The horizontal axis is time, and the vertical (density) axis is the probability that the species is detected at that time point. The integral value of the area under the curve is 1 [41]. This method is considered to have great advantages in studying animal activity rhythms based on passive shooting conditions [42]. Daily temporal overlaps were transformed from hours to degrees. Pairwise comparisons of the two deer were performed by estimating the coefficients of overlaps (Δ), and the estimator Δ4 was used as our sample sizes were all >75. The coefficients of overlaps ranged from 0 (no overlap) to 1 (complete overlap) [43]. These activity pattern analyses were performed by using the “Overlap” package of R [44]. Statistical significance was set at *p* < 0.05 [45].

## 3. Results

### 3.1. General Summary

A total of 235 and 153 valid presence sites of tufted deer and sambar were obtained for model construction, and the independent detections of tufted deer and sambar were 10,437 and 5203, respectively (Figure 2).

### 3.2. Spatial Niche

#### 3.2.1. Suitable Habitats

The MaxEnt model generated habitat suitability prediction for tufted deer with training AUC = 0.963 and test AUC = 0.955, while the training and test AUC for sambar equaled to 0.981 and 0.971, respectively, indicating a quite satisfactory model performances. The suitable habitats for tufted deer and sambar were predicted as 1038.40 km^2^ and 692.67 km^2^, accounting for 25.38% and 16.93% of the total area of the reserve, respectively (Figure 3).

#### 3.2.2. Important Environmental Variables

The results showed that the dominant contributing environmental variables of tufted deer were precipitation seasonality, vegetation types, NDVI, mean temperature of the driest quarter, total annual precipitation, and isothermality, and the dominant variables of sambar were precipitation seasonality, vegetation types, isothermality, mean temperature of the driest quarter, total annual precipitation, precipitation seasonality, and altitude. Furthermore, precipitation seasonality, vegetation types, mean temperature of the driest quarter, and total annual precipitation were the four common variables influencing the habitat selection of the studied species (Figure 4).

#### 3.2.3. Spatial Overlaps

The niche overlap indexes D and I between tufted deer and sambar were 0.703 and 0.930, respectively, indicating that the two species have relatively similar ecological needs and high overlap in geographical distribution. The overlapping area of suitable habitats for tufted deer and sambar was 656.67 km^2^, accounting for 63.24% and 94.80% of their respective total suitable habitat areas, respectively. The two species shared suitable habitats in the lower-elevation valleys of the reserve, whereas the tufted deer were mainly distributed on the periphery of these places.

### 3.3. Temporal Niche

#### Daily Activity Rhythm Characteristics and Overlapping Relationships

Both tufted deer and sambar exhibited activity peaks at dawn and dusk. The tufted deer’s activities peaked around 07:30–09:30 and 18:00–20:00. The activity intensity during the day was higher than at night; meanwhile, the activity peak in sambar was approximately during 06:00–08:00 and 19:00–21:00. The weakest activity intensity of sambar appeared around 12:00, and the activity intensity of sambar was greater at night than day (Figure 5a). The annual Δ4 of tufted deer and sambar was 0.781 (Wald test, *p* = 0.129), indicating that the annual difference in the daily temporal patterns of the two species was not significant.

Furthermore, during the dry season, the daily activity peak in tufted deer was during 08:00–10:00 and 17:30–19:30, while sambar were active during 06:30–08:30 and 18:00–20:00 (Figure 5b). During the rainy season, the daily activity peak in tufted deer was during 06:30–08:30 and 18:30–20:30, while sambar were active during 05:30–07:30 and 19:00–21:00 (Figure 5c) (rainy season: longer days and shorter nights in the northern hemisphere). The seasonal Δ4 of tufted deer and sambar in the dry and rainy seasons was 0.756 (Wald test, *p* < 0.001) and 0.767 (Wald test, *p* = 0.013), respectively, indicating that the seasonal difference in the daily temporal patterns of the two species was significant.

## 4. Discussion

Based on the 9-year camera-trap data in the Gongga Mountain National Nature Reserve, our study explored the spatio-temporal distribution patterns of the tufted deer and sambar. We found that the spatial niches of tufted deer and sambar show a high degree of overlap, their daily activity rhythms on the annual time scale also show extremely high similarity, and the temporal niches on the annual time scale do not have significant differentiation. However, on a more subtle time scale, i.e., on different seasonal time scales, the daily activity rhythm of the tufted deer and sambar show significant differences, indicating that the two species still have temporal niche separation under specific living conditions, which may be one of the mechanisms for tufted deer and sambar to achieve long-term stable coexistence in the reserve. This finding is similar to the results of existing studies on the coexistence patterns of other sympatric relative species [46,47], indicating that in the study of the coexistence mechanism of species, not only should we compare the niche differences between species from different ecological dimensions, but the influence of different spatio-temporal research scales on the research results should also be considered, which plays an important role in revealing the coexistence patterns of sympatric relative species and community stability mechanisms more scientifically [48].

For their spatial niches, the result of the MaxEnt model indicated that the contribution rates of the vegetation types for tufted deer (18.0%) and sambar (20.7%) were some of the highest. Furthermore, the response curves revealed that both species preferred to choose coniferous forests over mixed and broadleaved forests but exhibited no obvious selectivity for others. Both species showed a similar ecological need, and tufted deer showed a broader adaptation to precipitation seasonality, vegetation types, mean temperature of the driest quarter, and total annual precipitation, which explained why the two species had large, overlapping habitats and why the habitats of tufted deer were bigger. For their temporal niches, the results of kernel density estimation analysis revealed that the activity peaks in both species occurred at dawn and dusk; this is consistent with the previous studies showing that most wild ungulates are relatively active around sunrise and sunset [49,50,51]. Nevertheless, tufted deer maintained high activity intensity in the period after sunrise and before sunset, and the activity intensity in the daytime was higher than that at night. Meanwhile, the sambar mainly maintained a high activity intensity in the period after sunset and before sunrise, and the activity intensity at night was higher than that in the daytime. The activity patterns of both species changed with the changes in the photoperiod seasonally; the activity time of tufted deer increased with the length of the day, while for sambar, it was the opposite.

The habitat selection and behavioral rhythms of wild animals are the result of the comprehensive effects of many factors, including natural environmental conditions, biological–environmental conditions, and species physiological characteristics [52,53,54,55]. The results of this study revealed that the two species had a high degree of overlap in the spatial niche but a large degree of separation in the temporal niche, which is also consistent with the fact that the two species tended to appear in the same camera-trap region at different time periods within the same day. Both species are plant-eating herbivores, and their food composition changes with the change in plant abundance. The well-developed vegetation in the reserve provides sufficient food resources and good hiding conditions for the two species. Sufficient food resources and separation in temporal niches may be the reasons for the two species’ coexistence. 

The clarification of various ecological relationships and coexistence mechanisms between sympatric species must rely on the data of the spatio-temporal distribution of species on large rational scales and long timescales [48,56,57]. Some large- and medium-sized, terrestrial, wild animals often have the characteristics of low population density, strong migration ability, and sensitivity to human activities. Therefore, it is difficult to obtain their spatio-temporal distribution data using traditional methods [58,59]. The characteristics of the camera trap have the advantage of easy standardization and popularization, and the camera trap is increasingly recognized by many scientific research units and nature reserves at all levels. The camera trap has been widely used in wildlife protection and plays an indispensable role in policy and evaluation of the effectiveness of reserve management [60,61,62,63]. Since the implementation of the camera-trap project in the reserve in 2012, the monitoring results have provided data support for species protection management and related scientific research [26,64]. However, there are still deficiencies in the monitoring area and efforts [65,66]. In addition, it is worth noting that the use of camera traps has some non-negligible limitations in studying the spatio-temporal distribution of wildlife. This study assumed that the activity density of the target species was proportional to the captured rate of the camera traps and the differences in the movement speed, body size, appearance, and interaction of accessing the same infrared camera between tufted deer and sambar. After all, these may cause differences between the observed and the real results of distribution and activity rhythm [40,67,68,69].

## 5. Conclusions

Our study found that the coexistence of tufted deer and sambar was mainly due to their differentiation in the timing of activity periods rather than their differentiation of habitat use and population distribution. The suitable habitat area of sambar in the reserve is much smaller than that of tufted deer, and the sambar habitat is mainly located within the suitable habitat of tufted deer, indicating that sambar may face a greater population survival crisis. Maintaining and protecting the current ecological environment of the two species’ common suitable habitat is of great significance to the population continuation of the two species in the reserve. In addition, precipitation seasonality and vegetation conditions are the common main environmental factors affecting the habitat suitability of the two species. Therefore, the monitoring of vegetation growth conditions in the suitable habitats of the two species should be strengthened, and the two species should be strengthened in years with extreme precipitation. Monitoring habitat suitability is important so that appropriate conservation measures can be taken. At the same time, after the habitat suitability of the two species decreases, the monitoring of the activity rhythm of the two species should also be strengthened to clarify the adjustment mode of the activity rhythm of the two species under the condition of insufficient living resources and whether the adjusted activity rhythm of the two species will intensify interspecific competition, thereby intensifying the survival crisis of vulnerable, competing species populations. 

## Figures and Tables

**Figure 1 animals-12-02694-f001:**
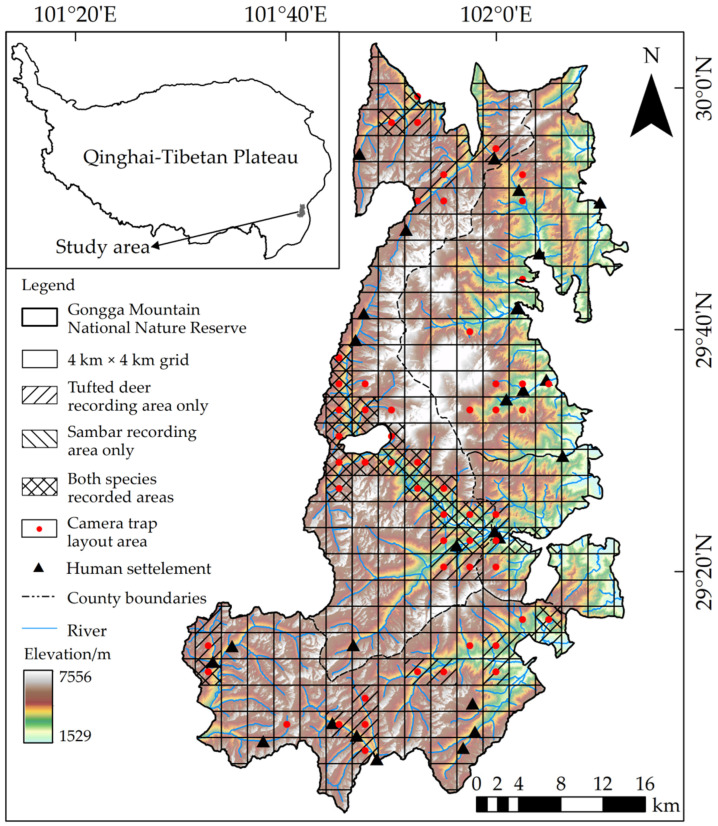
Distribution grid records for tufted deer and sambar recorded in the Gongga Mountain National Nature Reserve, China.

**Figure 2 animals-12-02694-f002:**
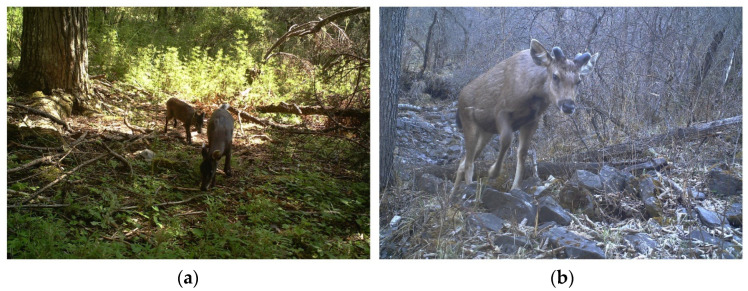
Photos of tufted deer (**a**) and sambar (**b**) in the Gongga Mountain National Nature Reserve, China.

**Figure 3 animals-12-02694-f003:**
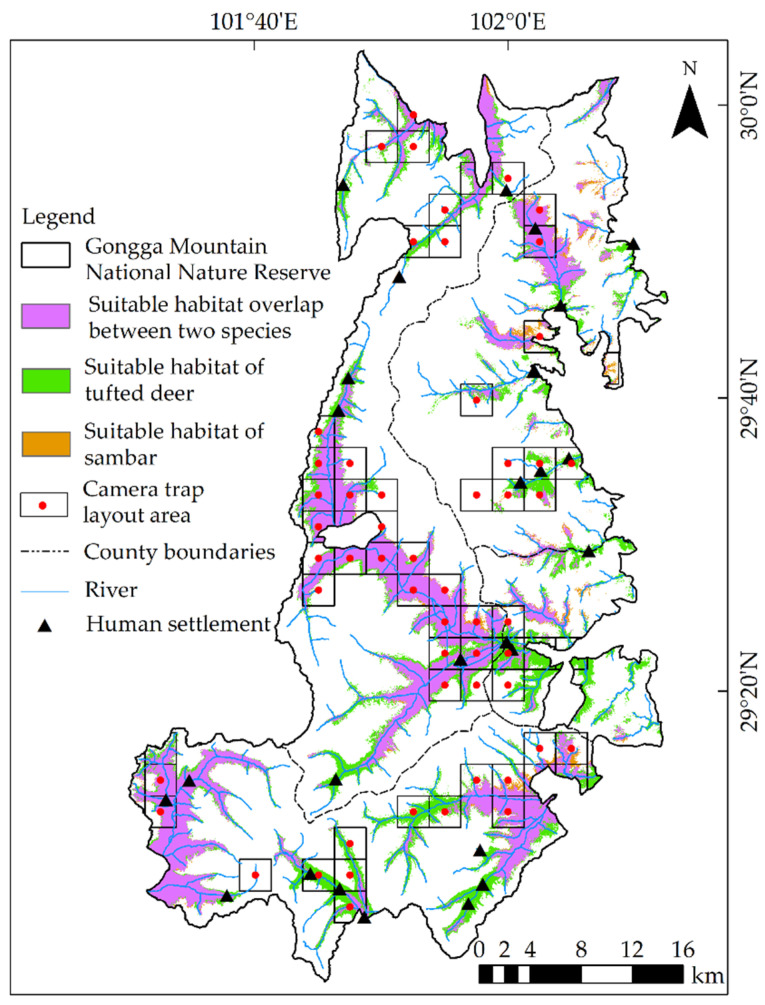
Distribution of tufted deer and sambar habitats in the Gongga Mountain National Nature Reserve, China.

**Figure 4 animals-12-02694-f004:**
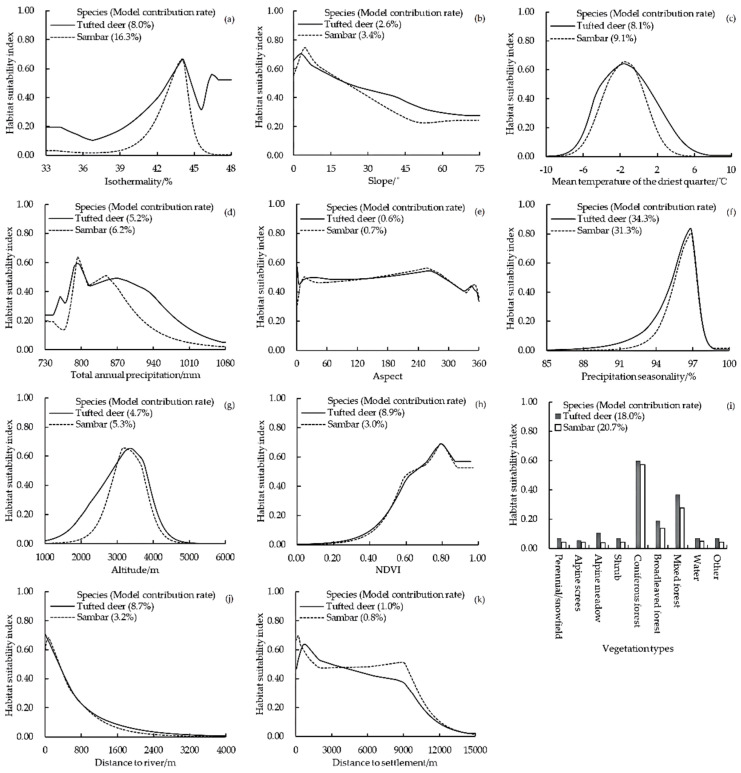
Response curves of habitat suitability index of tufted deer and sambar against the 11 environmental variables (**a**–**k**), with percent contribution of each variable for each species shown in following parentheses.

**Figure 5 animals-12-02694-f005:**
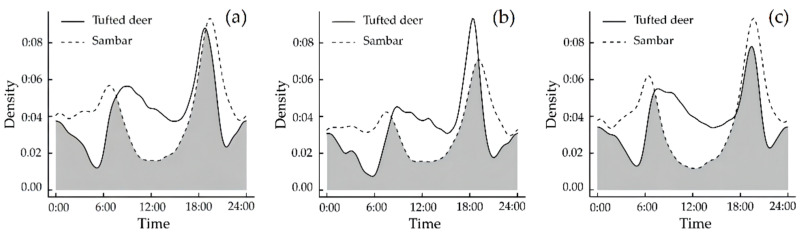
The daily activity patterns of tufted deer and sambar in the annual time scale (**a**), dry season time scale (**b**), and rainy season time scale (**c**). Overlaps are represented by the shaded gray area.

**Table 1 animals-12-02694-t001:** Environmental factors involved in model operation and their sources.

Classification	Variable	Source
Topographic variables	Elevation	http://www.gscloud.cn (accessed on 15 April 2022).
	Slope	
	Aspect	
Vegetation variables	Vegetation type	https://data.ess.tsinghua.edu.cn (accessed on 10 March 2022)
	Normalized difference vegetation index (NDVI)	http://www.nesdc.org.cn (accessed on 13 April 2022)
Climate variables	Isothermality	https://worldclim.org (accessed on 25 February 2022)
	Mean temperature of the driest quarter	
	Total annual precipitation	
	Precipitation seasonality	
Water source distance	Distance to river	http://loess.geodata.cn (accessed on 26 March 2022)
Human disturbance variable	Distance to human settlement	

## Data Availability

The datasets used in this study are available from the corresponding author on reasonable request.

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
