# Peer review of "Spatio-Temporal Niche of Sympatric Tufted Deer (Elaphodus cephalophus) and Sambar (Rusa unicolor) Based on Camera Traps in the Gongga Mountain National Nature Reserve, China"

_animals, 2022, doi:10.3390/ani12192694_

Round 1
Reviewer 1 Report
Dear Authors,
I have some suggestions to your paper, but in general in my opinion it is a valuable work.
I will clarify my remarks line by line:
Line 5 - fringe - please discuss it with a native speaker - in my opinion it is not a correct word in taht meaning. Maybe corner? Maybe edge? Please change it in whole text because there are repetitions of this word.
Line 10 - 3School - please add a break between
8, 9, 11 - please look at signs in the ends of lines - ; or " - please standardize it.
20 - ....on 10-year (2012–2021) - did you start in January 2012 and finished in December 2021? if not, it is not 10 years period.
22 - reserve, not rerserve
18, 19, 22, etc - we write name of the species in small letters in a middle of sentence, like: "The tufted deer..." - change it in whole text.
26 - nd the integrated management - delete the
29 - add something on the begining of a sentence - before (1)
30 - deers
45 - it is a beginning sentence and something is wrong here.
Maybe: Niche includes temporal, spatial and feeding niche.
Ask a native speakers, word should not be doubled in one sentence.
45, 47 - not [1-2] or [3-4] but [1,2] and [3,4] - do you see a difference? e.g. [5-8] is correct. Please change in whole text.
63 - red deer, roe deer - small letters
75-76 - this sentence doesn not match here. Please delete it.
77 - delete (2012–2021) - you will write it in Methods
128 - <0.75[31]; - put a break after 0.75
133 - ArcGIS - please clarify which number
136 - resolution without '-' - just 30 m
179 - each partial figure should be called A, B, C. In truth B - detailed China it is not needed because you could put a blue studied area in figure A. In figure C I would change a colour of camera traps - maybe yellow? Black ones are not so good visible.
183 - just Photos, not The photos
185 -till this moment you put everywhere dots, e.g. 3.1., 3.1.1., not without them. Please standardize it.
193 - habitats
204 - 'of the two' - maybe of studied species?
221 - maenwhile - small letter or is it a new sentence? the same in 243 line.
236-238 - please standardize what you put in the end of the sentence under figures. A dot or nothing? Before it you did not put dots.
240 - please delete it: (2012–2021) - we know it already.
247-248 - a discussion chapter is a different part then results. You should not repeat information like:
| 656.67 km2, which accounted for 63.24% and 94.80% of | 247 |
| the total area of Tufted and Sambar deer, |
299-302 - these are results, not conclusions.
305-309 - these two sentence are much similar to lines 78-82.
You should work more in Conclusions chapter, remember it is a separate chapter, not repetition of results or discussion. It is diffucult, because authors should be open-minded to write Conclusions in a proper why, but you will handle it, I'm sure. Good luck.
311-316 - two big breaks after ';' and next activity, like
...; investigation...
320 - till the end - each journal name should be written in its shorter name. Add DOI where it is possible.
Reviewer 2 Report
This work reports the spatio-temporal patterns of two sympatric deer species in a reserve in China. The authors provided data from camera-traps deployed for 10 years in a vast area, which has a huge merit. However, the methodology is not well explained, and more details should be added to properly evaluate the scientific quality of the work. Therefore, I cannot accept the manuscript in the current form. Please, find my comments below and in the attached file.
Novelty and scientific interest: It is not clear why to study the spatio-temporal of these two species is important. Why your paper is interesting for the international readers? Why did the authors select these two species? You should convince to the readers in the introduction.
Camera-traps: More information concerning camera-traps is necessary. How many days were they deployed at each site?, Did you use some baits or attractants? Which camera models did you use? The distance between the cameras, the overall sampling effort (cameras*nights), the altitude range in which the cameras were deployed, and more details about the setup options: photos or video, interval, 3 consecutive photos…
Environmental data and data processing: A table with all the variables and their description is necessary. It is not clear which variables (and how many) they use.
Discussion: The use of camera-trap to study spatio-temporal trends have some important limits. In fact, you assumed that the activity density of the target species was proportional to the captured rate of the camera traps. However, there are other factors that also affect the detection probability by camera-traps, such as the size of target species, the behaviour, the home-range… The authors should include a paragraph in the discussion to recognize these limitations of the study. Here you have some useful references:
Broadley K., Burton A.C., Avgar T. & Boutin S. 2019. Density‐dependent space use affects interpretation of camera trap detection rates. Ecol. Evol. 9: 14031–14041.
Sollmann R., Mohamed A., Samejima H. & Wilting A. 2013. Risky business or simple solution - Relative abundance indices from camera-trapping. Biol. Conserv. 159: 405–412.
Tourani M., Brøste E.N., Bakken S., Odden J. & Bischof R. 2020. Sooner, closer, or longer: detectability of mesocarnivores at camera traps. J. Zool. 312: 259–270.
Conclusion: The conservation and management implications of the results are not clear. The authors should explain why this study is important for the conservation and management of both species. For instance, in the summary they said that the results provide scientific information for effective wild animal conservation in the reserve, but they do not propose any specific measure that can be applied according to their results.

Round 2
Reviewer 2 Report
The authors have considered all my previous suggestions, so the manuscript can be accepted in the present form.